# Quantifying Extreme Precipitation Forecasting Skill in High-Resolution Models Using Spatial Patterns: A Case Study of the 2016 and 2018 Ellicott City Floods

**Stephanie E. Zick** 

Department of Geography, Virginia Polytechnic Institute and State University, Blacksburg, VA 24060, USA; sezick@vt.edu; Tel.: +1-540-231-3434

**Abstract:** Recent historic floods in Ellicott City, MD, on 30 July 2016 and 27 May 2018 provide stark examples of the types of floods that are expected to become more frequent due to urbanization and climate change. Given the profound impacts associated with flood disasters, it is crucial to evaluate the capability of state-of-the-art weather models in predicting these hydrometeorological events. This study utilizes an object-based approach to evaluate short range (<12 h) hourly forecast precipitation from the High-Resolution Rapid Refresh (HRRR) versus observations from the National Centers for Environmental Prediction (NCEP) Stage IV precipitation analysis. For both datasets, a binary precipitation field is delineated using thresholds that span trace to extreme precipitation rates. Next, spatial metrics of area, perimeter, solidity, elongation, and fragmentation, as well as centroid positions for the forecast and observed fields are calculated. A Mann–Whitney *U*-test reveals biases (using a confidence level of 90%) related to the spatial attributes and locations of model forecast precipitation. Results indicate that traditional pixel-based precipitation verification metrics are limited in their ability to quantify and characterize model skill. In contrast, an object-based methodology offers encouraging results in that the HRRR can skillfully predict the extreme precipitation rates that are anticipated with anthropogenic climate change. Yet, there is still room for improvement, since model forecasts of extreme convective rainfall tend to be slightly too numerous and fragmented compared with observations. Lastly, results are sensitive to the HRRR model's representation of synoptic-scale and mesoscale processes. Therefore, detailed surface analyses and an "ingredients-based" approach should remain central to the process of forecasting excessive rainfall.

**Keywords:** hydrometeorology; spatial analysis; precipitation verification; extreme flood

## 1. Introduction

Of all the hydrometeorological hazards, floods are the leading cause of fatalities in the United States, based on a National Weather Service (NWS) assessment of 30 years of weather-related fatalities from 1988 to 2017 (http://www.nws.noaa.gov/os/hazstats.shtml). This ranks floods above tornadoes, derechos, hailstorms, and lightning. Notably, floods not only threaten human lives but also cause extensive damage to property and infrastructure [1], destruction to crops and loss of livestock [2], and deterioration of health conditions [3]. Due to the profound impact of floods on lives and livelihood, it is crucial to understand and forecast precipitation events with sufficient skill to inform the public and stakeholders about expected risks of flood.

As a result of climate change and urbanization, flood statistics are non-stationary [4]. Theoretical, modeling, and observational research suggests that rainfall rates are increasing [5–7] and extreme rainfall events are becoming more frequent [8–10]. The most recent report from Intergovernmental Panel on Climate Change (IPCC) finds with high confidence that precipitation over mid-latitude

land areas in the Northern Hemisphere has increased since 1951 and predicts that it is "very likely" that extreme precipitation events over mid-latitude land masses will become more intense and more frequent by the end of the 21st century [11]. Urbanization [12–14] and orographic enhancement [15] may play a critical role in the increased incidence of flooding in certain regions, as well as an increased occurrence of long duration flooding events from both tropical and extratropical cyclones [16,17].

Recent historic floods in Ellicott City, MD, on 30 July 2016 and 27 May 2018 provide stark examples of the types of floods that are expected to become more frequent due to urbanization and climate change. In both cases, rainfall rates exceeded two inches (>50 mm) per hour, leading to an overabundance of run-off, which taxed the city's storm water infrastructure. The meteorological environments which heralded these two precipitation events were complex, involving the interaction of multiple scales of atmospheric processes that ultimately focused extreme precipitation over Ellicott City and other nearby municipalities to the west of Baltimore, MD. This meteorological set-up resulted in catastrophic rainfall amounts in a short period of time. For a numerical weather prediction (NWP) model to predict these types of events accurately, it must be able to handle dynamical and physical processes over multiple space and time scales. For instance, the synoptic-scale pattern must accurately portray large scale wind and moisture fields in order to forecast the moisture convergence pattern. Frontal and near surface boundaries, which can act as forcings for localized ascent, must also be captured reasonably. At more localized scales, convective and microphysical parameterization schemes must work in conjunction to produce precipitation via sub-grid-scale processes.

Given the expectation of more frequent extreme rainfall events, it is crucial to evaluate the capability of our state-of-the-art NWP models in predicting hydrometeorological events such as the Ellicott City floods. Specifically, to forecast flash flooding, meteorologists (and the model forecasting tools upon which they rely) must be able to predict the evolution of convective-scale (horizontal length scales <10 km) storm elements, which account for the highest rainfall rates [18]. Additionally, the models need to capture synoptic-scale (horizontal length scales >1000 km) and mesoscale (horizontal length scales between 10 and 1000 km), features that result in training cells or that lead to the development of large, nearly stationary precipitation complexes. Current high-resolution convection-allowing models (CAMs), such as the High-Resolution Rapid Refresh (HRRR) model were developed to capture convective-scale weather phenomena as well as their interaction with larger-scale forcings. In fact, the original release notes from the National Centers for Environmental Prediction (NCEP) stated, "The HRRR provides forecasts, in high detail, of critical weather events such as severe thunderstorms, flash flooding, and localized bands of heavy winter precipitation" [19]. Yet, as of 2018, only a few studies [20–23] have investigated its skill in representing the mesoscale and convective processes that lead to flash flooding.

While there is a lack of research into the specific performance of CAMs in representing extreme precipitation events across multiple scales, there has been abundant research more generally into model skill in predicting precipitation, better known as precipitation verification. Model verification studies provide important information about forecast quality and systematic errors or biases. These studies help end-users in their interpretation of model forecast products and also direct the efforts of researchers in the development of model improvements. Traditional precipitation verification generally fits into one of two categories: visual verification or pixel-based methods. A pixel-based method involves a grid-to-grid comparison between model forecasts and observations from radar, gauge, satellite, or some combination of the above. Using pixel-based methods, users can evaluate summary statistics (e.g., correlations and root-mean-square-error), statistics based on contingency tables (hits, misses, false alarms, and correct negatives) and statistics that derive from contingency tables (probability of detection and various skill scores) (see Wilks [24] for a comprehensive review).

In recent years, with the rise of high-resolution (horizontal grid spacings ≤4 km) NWP models, there has been increasing interest in new spatial methodologies for precipitation verification. This interest has arisen out of a need to characterize precipitation forecast skill when traditional metrics are hindered by a "double penalty" associated with errors in the timing and/or location of a given

precipitation feature [25,26]. Several spatial methodologies have been developed. In this study, an object-based approach is utilized, an approach that has become increasingly popular in the research and operational meteorology communities since the development of Contiguous Rain Area [27] and Method for Object-Based Diagnostic Evaluation [28]. In an object-based approach, precipitation objects, or polygons, are first identified by applying a threshold value to create a binary precipitation field. The polygons are then evaluated for their spatial attributes, which can include location, two-dimensional measures of shape, and average rainfall intensity. Such an objective method is ideal for the present study, since it circumvents the double-penalty problem and also allows for an evaluation of biases with respect to the shape and spatial attributes of the forecast precipitation versus observations.

Due to the nonlinear nature of atmospheric convection, determining an appropriate threshold for delineating a binary precipitation field is a nontrivial subject. In some applications, an ideal method is to delimit a region linked to the fundamental dynamical and physical mechanisms that produce and organize precipitation. For example, Matyas [29] examined strict 30–40 dBZ reflectivity thresholds (approximately 10 mm h$^{-1}$) as the empirically-based separation between stratiform and convective precipitation. However, a lower or higher threshold may be more meaningful to hydrologists and/or emergency managers. Since object-based methods for precipitation verification are sensitive to the threshold value [30–32], it is important to evaluate the sensitivity of model forecast skill to the threshold that is specified.

This study utilizes an object-based approach to precipitation verification to quantify HRRR model forecast skill in predicting rainfall in the two aforementioned events that led to catastrophic flooding in Ellicott City and surrounding regions. Two research questions are addressed: (1) can object-based methods offer insight over traditional precipitation verification methods in evaluating NWP skill in predicting these extreme rainfall events? and (2) how does forecast skill vary over a range of rain rate thresholds from "trace" to moderate to extreme precipitation rates? These research questions are evaluated as a proof of concept, and therefore it is important to contextualize the results based on the synoptic and mesoscale environments associated with each event, so as to inform future, more comprehensive studies into high-resolution model precipitation forecast skill.

## 2. Materials and Methods

### 2.1. Case Studies

In this section, a brief overview is provided for each convective event to aid with interpretation of the results. In both events, catastrophic flooding occurred in the Ellicott City area. Heavy rainfall and urbanization provided ideal conditions for flash flooding in both cases. However, there are a few important distinctions between characteristics of the large-scale environment that promoted the development of extreme precipitation in these two events.

#### 2.1.1. Case Study 1: 30 July 2016

Intense convective rainfall led to catastrophic flooding in historical Ellicott City, MD, on the evening of 30 July 2016. In total, nearly 150 mm of rain fell over a narrow swath of central Maryland (Figure 1a) to the west of Baltimore. Rain gauge data from Ellicott City support these rainfall totals. During the flood, city streets became raging rivers in the old town of Ellicott City. Two fatalities and significant properties losses were reported [33].

Prior to the event, there was limited advance warning. The first public alert was released as a flash flood warning at 19:17 local time on 30 July 2016, and the first report of flooding was recorded at 20:01 in Chatham, MD, to the east of Ellicott City. According the NWS, the majority of the severe flooding occurred between 20:05 and 21:00 [33]. Although there was a little advance warning, in terms of public watches, warnings and advisories, this event was also not a complete surprise, as severe thunderstorms with locally heavy rainfall were forecast to occur by both the NWS and local broadcast meteorologists. Additionally, the Weather Prediction Center (WPC) released a Mesoscale Precipitation Discussion

(https://www.wpc.ncep.noaa.gov/metwatch/metwatch_mpd_multi.php?md=0506&yr=2016) at 17:00 local time.

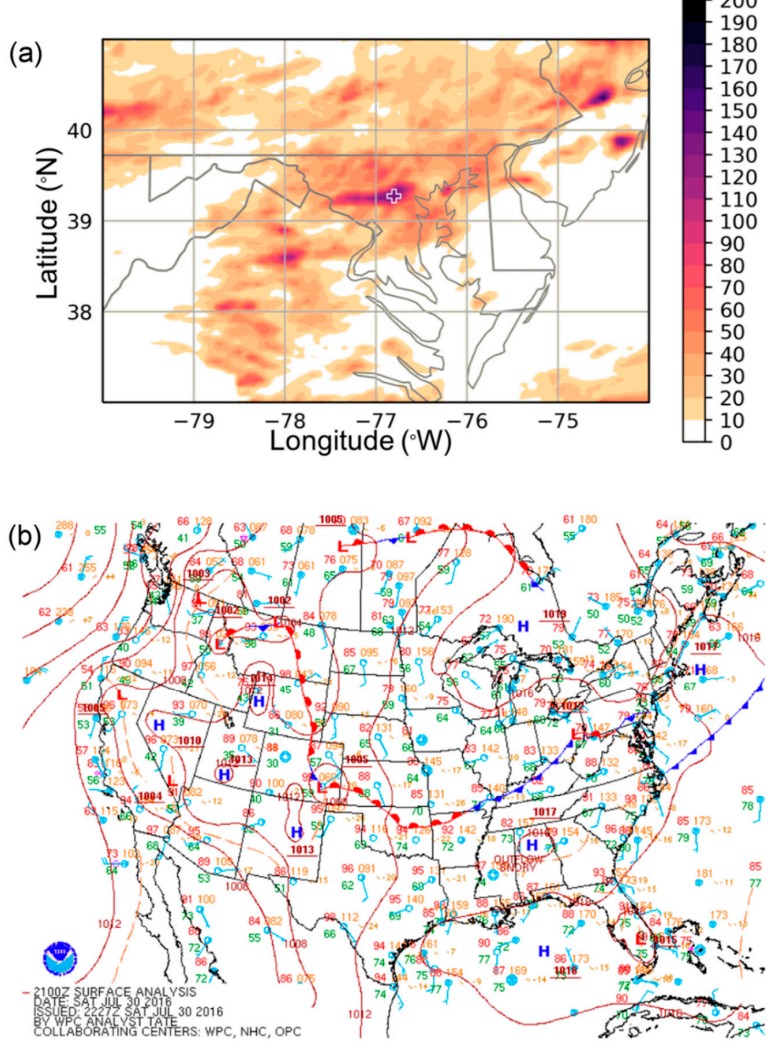

**Figure 1.** Meteorological overview of the July 2016 flood. (**a**) Stage IV estimated rainfall totals for the 24 h period ending 12 UTC 31 July 2016 and (**b**) National Weather Service (NWS) Weather Prediction Center surface maps for the continental United States valid at 21 UTC 30 July 2016. In the top panel, the location of Ellicott City is indicated by a plus sign near the center of the domain at 39.27°N, 76.80°W.

The large-scale environment (Figure 1b) featured low pressure over southeastern Ohio and ridging over the subtropical western Atlantic. A stalled frontal boundary was situated along the Pennsylvania-Maryland border, and a weak high also developed off the coast of New England. This synoptic pattern led to south-southwesterly low-level flow over the mid-Atlantic region and anomalously high precipitable water values.

The rainfall occurred over a fairly short time interval, beginning at 18:00 local time (22:00 UTC) and ending at 21:00 local time (01:00 UTC). Interestingly, this rainfall was associated with both training and back-building convection. First, several weaker convective elements developed and traveled from south to north as a series of training cells that moved directly over Ellicott City. Subsequently, a line of north-south oriented convection moved in from the west and interacted with these weaker convective elements. The second north-south oriented line continued to produce new convection through a process known as backbuilding. The heaviest rains occurred between 2330 UTC 30 July and 0030 UTC 31 July [33].

2.1.2. Case Study 2: 27 May 2018

On the afternoon and evening of 27 May 2018, just 22 months after the 2016 floods, a second torrential rainfall event occurred in the Ellicott City area. Precipitation totals for the 24 h period ending 12 UTC 28 May (Figure 2a) show much more isolated rainfall compared with the 2016 event. Radar estimates (not shown) indicate that maximum rainfall totaled over 220 mm (9 in) along a corridor from Ellicott City to Catonsville, which is a distance of approximately 6 km. Since the 4 km Stage IV dataset is similar in horizontal grid spacing to the scale of the maximum precipitation corridor, it may be unable to resolve these maximum rainfall totals. Still, a clear precipitation maximum exceeding 120 mm can be seen in the 24 h rainfall totals (Figure 2a). This highly localized extreme precipitation near Ellicott City led to flash floods that were very devastating, causing one fatality as well as catastrophic damage to buildings, vehicles, and roads in and around the city [34].

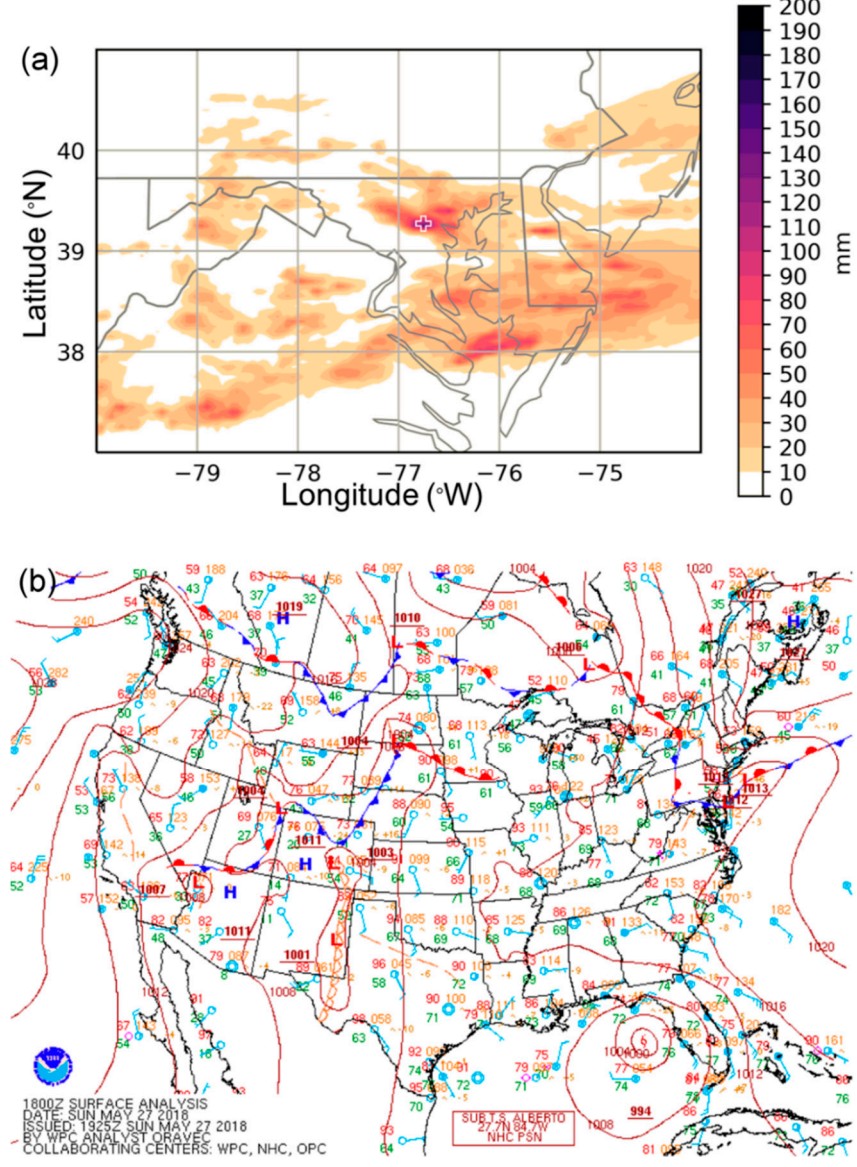

**Figure 2.** Meteorological overview of the May 2018 flood. (a) Stage IV estimated rainfall totals for the 24 h period ending 12 UTC 28 May 2018 and (b) NWS Weather Prediction Center surface maps for the continental United States valid at 18 UTC 27 May 2018. In the top panel, the location of Ellicott City is indicated by a plus sign near the center of the domain at 39.27°N, 76.80°W.

In contrast with the 2016 floods, this second flooding event featured multiple advance alerts in the days and hours preceding the floods. The NWS first alerted the public about potential flooding in a Hazardous Weather Outlook on Friday, 25 May. By Sunday morning on the day of the event, the NWS ascribed sufficient confidence in the potential for damaging flooding to issue a Flood Watch for the Baltimore/Washington region, with an expectation of heavy rain and flooding during the afternoon and evening hours. As the rain began that afternoon, a Flood Warning was issued at 15:19 local time for "portions of Howard County, Baltimore County, and Baltimore City including Ellicott City, Catonsville, Dundalk, and Baltimore City" [34].

The synoptic environment (Figure 2b), as shown by the surface map valid at 18 UTC on 27 May, revealed a low pressure system over southern Ontario with a warm front extending east and southeast toward the northeastern United States. Near Ellicott City, this warm front was nearly stationary, draped along the Pennsylvania-Maryland border, just to the north of the city. In fact, just prior to the onset of precipitation, NWS forecasters observed the progression of a back door cold front toward northeastern Maryland. Another important feature on this day was Subtropical Storm Alberto, over the Gulf of Mexico just off the coast of Florida. Cyclonic circulation around the subtropical storm led to strong southeasterly flow off the warm Gulf stream waters toward the mid-Atlantic states, providing ample, deep moisture to the region. Precipitable water values exceeded 50 mm across the tri-state region of Delaware, Maryland and Virginia. A broad shortwave trough over southeastern Pennsylvania and daytime heating also contributed to the triggering of convection in the afternoon on 27 May 2018 [34].

In contrast with the 2016 event, the 27 May 2018 precipitation occurred in two waves with one-hour rainfall rates in each wave exceeding 50 mm $h^{-1}$ (2 in $h^{-1}$) as measured by the Ellicott City rain gauge. The first wave occurred at 15:20–16:20 local time (19:20–20:20 UTC) and the second wave at 17:00–18:00 local time (21:00–22:00 UTC). Between these two waves, there was a brief period of lighter precipitation. NWS forecasters identified multiple low-level boundaries that acted as lifting mechanisms to initiate the convection associated with these cells. Once initiated, the extremely moist synoptic-scale environment contributed to the extreme rain rates that were observed in these cells [34].

*2.2. Data*

2.2.1. HRRR

The HRRR model is part of NCEP's suite of "rapid refresh" products, which are updated at hourly intervals to provide forecast guidance for aviation and short-range forecasting. The HRRR model is nested in the 13 km resolution Rapid Refresh (RAP) model, which provides initial and lateral boundary conditions. With approximately 3 km grid spacing and 50 vertical levels, the HRRR model is categorized as a cloud resolving, convection allowing-model (CAM), and it is built using the Advanced Research Weather Research and Forecasting (WRF-ARW) model infrastructure. A unique, innovative component of the model is the assimilation of radar data from the NWS's Weather Surveillance Radar (WSR)-88D radar network [19].

In this study, the operational HRRR model output that was made available to weather forecasters at the time of the event is utilized. These data are archived and maintained at the University of Utah's online archive (doi:10.7278/S5JQ0Z5B) for model output beginning 15 July 2016. Documentation on this online archive is available in Blaylock and others [35]. As an operational model, the HRRR is frequently upgraded to improve its algorithms and overall performance. Therefore, the currently operational version of the HRRR does not fully reflect the versions that were operational in 2016 and 2018 at the times of the Ellicott City case studies. In fact, a significant upgrade occurred between the two case studies: HRRRv1 was operational in July 2016 and HRRRv2 was operational in May 2018. Important upgrades associated with version 2 are summarized in Table 1 and include transition to a newer version of WRF-ARW; significant updates to microphysics, planetary boundary layer, and land surface model parameterization schemes; and new assimilation of radial wind, surface mesonet, and satellite radiance data. These upgrades were intended to address known biases in

HRRRv1, including a daytime warm/dry bias in the warm season that could negatively impact convective forecasts [19]. Since significant model upgrades can lead to detectable improvements in the representation of convection [36], the results in this study need to be considered in that context.

**Table 1.** High-Resolution Rapid Refresh (HRRR) model version comparison.

| Model Specification | HRRRv1 | HRRRv2 |
|---|---|---|
| Operational implementation | 30 Sep 2014 | 23 Aug 2016 |
| Weather Research and Forecasting (WRF) model version | WRF-ARW v3.4.1 | WRF-ARW v3.6.1 |
| Data assimilation system | GSI 3DVAR with assimilation of radar reflectivity | GSI Hybrid 3D-VAR/Ensemble with new assimilation of radial velocity, surface mesonet, and satellite radiance data |
| Initialization | RAP model | RAP model with new cycling of land surface states |
| Parent model (RAP) convective scheme | Grell 3-D | Grell-Freitas |
| Physical parameterization schemes | N/A | significant updates to microphysics, planetary boundary layer, and land surface model schemes |
| Forecast lead times | 0–15 h (every 15 min, archived hourly) | 0–18 h (every 15 min, archived hourly) |

### 2.2.2. Stage IV Precipitation Analysis

The NWS/NCEP Stage IV quantitative precipitation estimate is frequently used for precipitation intercomparison and verification studies [21,36–39]. Since 2002, this rainfall product has been generated in near-real-time by NCEP using the Next-Generation Radar (NEXRAD) Precipitation Processing System [40] and the NWS River Forecast Center (RFC)'s precipitation processing algorithm [41]. The dataset is available online from the NCAR Earth Observing Laboratory (doi:10.5065/D6PG1QDD).

The complete precipitation processing includes multiple steps. First, the data are preprocessed, converted to rain rates, and accumulated into hourly estimates [40]. Next, a multisensor precipitation estimate (MPE) is produced by merging radar and gauge data and applying a bias correction. A mosaic is then generated from multiple radars, and the precipitation estimates are gridded on the national Hydrological Rainfall Analysis Projection (HRAP) grid, which is polar stereographic map projection with approximately 4 km × 4 km grid spacing [42] over the continental United States (CONUS). Stage IV is the mosaicked final product from the 12 RFCs. NCEP produces this mosaic at hourly, 6 hourly, and 24 hourly intervals [43].

There are a few known issues and biases with the Stage IV product. First, the 24 h product is the most reliable and accurate product for post-analysis, as the daily data are reprocessed 24 h after the verification time with a manual quality control. Hourly and 6 hourly products are not manually quality controlled and occasionally include missing analyses from one or more RFC. Second, discontinuities can arise as a result of the mosaicking process due to hot and cold biases in individual radars as well as different rainfall algorithms applied by each RFC [44]. Therefore, while the Stage IV observations will be used as "ground truth" for the purpose of model verification, it is important to note that the observations are not perfect and may include some errors.

### 2.2.3. Data Processing

Prior to performing the object-based method, some data pre-processing is required, including important decisions about the time period and spatial domain over which the analysis is performed. The results are sensitive to both these considerations, and some care needs to be taken to select appropriate space and time scales [25,28,45]. In the present study, the study area is a regional domain centered on Ellicott City, MD, which spans 80°W to 74°W longitude and 37°N to 41°N latitude (Figures 1a, 2a and 3a,b). This domain encapsulates a region over which important mesoscale phenomena occurred and led to extreme precipitation. Therefore, this region is appropriate for

application of an object-based method for evaluating the skill of the HRRR model in representing the spatial characteristics of precipitation on the two days under consideration. For the temporal period, this analysis begins 10 h before onset of the observed peak precipitation rates and ends when the precipitation rates fall to zero. This time window was selected to make maximum use of available data, since the HRRR v1 model provided 15 h forecasts. For consistency, the same convention was used for the second case study, even though the HRRRv2 model upgrade extended the forecast lead time to 18 h (Table 1).

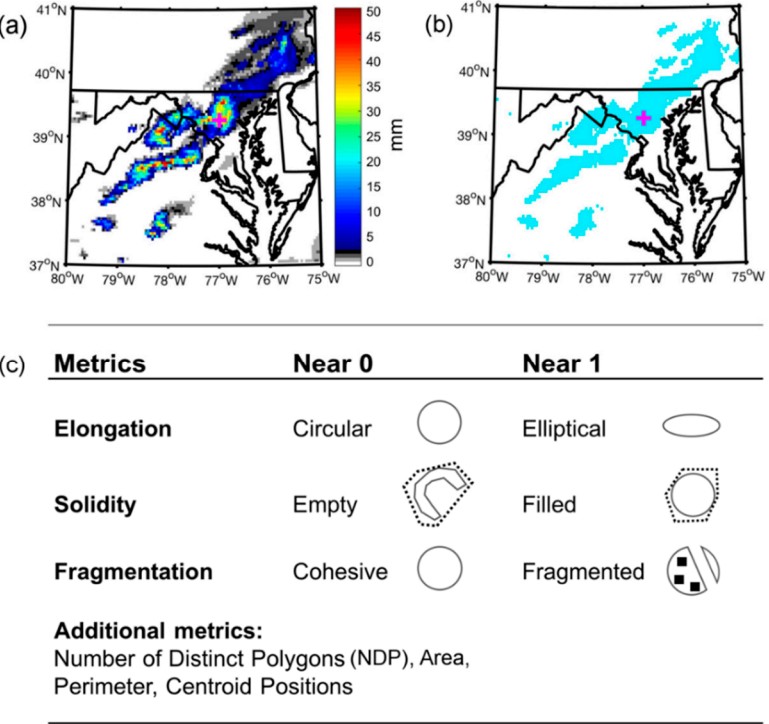

**Figure 3.** Demonstration of methodology for calculating object-based metrics based on (**a**) the full precipitation field, (**b**) the binary image corresponding to precipitation greater than 2.54 mm, and (**c**) spatial metrics included in study.

Additionally, an object-based verification is sensitive to the number of pixels and the horizontal grid spacing of the datasets under consideration [45]. Thus, a common grid for both datasets is needed. For their native grids, the Stage IV precipitation is provided on a polar stereographic grid with approximately 4 km grid spacing, while the HRRR model output is provided on a Lambert-conformal grid with 3 km grid spacing. For a common grid, a 0.05° latitude-longitude mesh is created for the study area. The domain is 99 × 81 and consists of 8019 pixels. Using a nearest neighbor interpolation, the precipitation is calculated for each dataset on this common grid. A bilinear interpolation was selected after comparison with other common interpolation schemes, specifically cubic spline and nearest neighbor. When interpolating the original data to the new latitude-longitude mesh and then back to the original grid, the nearest neighbor scheme resulted in the lowest root-mean-square errors (not shown). This simple analysis reveals that the choice of an interpolation scheme for precipitation studies is not trivial and should be investigated further (e.g., [46]).

*2.3. Spatial Metrics*

In this study, an object-based, spatial metric approach is utilized to evaluate precipitation forecasts from the HRRR model. As discussed in Section 1, precipitation verification methods can be categorized into a few overarching approaches: (1) visual verification, (2) pixel-based methods

(including correlations and dichotomous skills scores), and (3) object-based methods. A pixel-based method, the two-dimensional histogram, will be employed in this study to illustrate the merits and downfalls of such an approach in evaluating mesoscale precipitation signatures. As other authors have noted [27,28,45], pixel-based methods can provide valuable insight, but there is a notable "double penalty" in the event that a precipitation feature is forecast well but that precipitation feature is spatially displaced from that of the observed. Object-based measures have the capability of evaluating the precipitation feature by its shape, size, and other spatial attributes.

After the binary image is delineated, the spatial pattern of that image is quantified using three spatial metrics that characterize precipitation structure: solidity, fragmentation, and elongation (Figure 3c). Similar metrics have been applied in other object-based precipitation verification studies [30,45,47]. For simplicity and ease of comparison, all metrics have been devised to range from zero to one (Figure 3c). A short description of each metric follows.

### 2.3.1. Solidity

Solidity [48] is a measure of the fill of a precipitation object and ranges from empty (zero) to completely filled or "solid" (one). Formally, it is the ratio of the area of a particular binary object versus the area of its convex polygon, where a convex polygon represents the smallest polygon with acute angles that can encompass the precipitation object. In this study, the solidity is the summation of the solidity of all distinct precipitation objects in the domain:

$$Solidity = \sum_{i=1}^{NDP} \frac{Area_i}{Convex\ Arean_i} \qquad (1)$$

where NDP is the number of distinct polygons.

### 2.3.2. Fragmentation

Fragmentation [49] is an overall measure of characteristics of the precipitation distribution across the domain, with values near zero indicating a cohesive pattern and values near one indicating a fragmented pattern. It is the inverse product of solidity (Equation (1)) and a modified connectivity [45,49]:

$$F = 1 - \left[ \sum_{i=1}^{NDP} \frac{Area_i}{ConvexArea_i} \right] \left[ 1 - \frac{NDP - 1}{NDP + \log_{10} Area} \right] \qquad (2)$$

As such, fragmentation is correlated with solidity (r ~ 0.51), although it elucidates additional useful information about the precipitation distributions, as will be demonstrated in the results.

### 2.3.3. Elongation

Elongation provides information about the elliptical nature of the precipitation object, and ranges from circular (near zero) to elongated (near one). There are numerous ways to measure the elliptical nature of an object. In this study, elongation is calculated as the inverse of a circularity measure utilized by Stoddart [50] and, more recently, Matyas and others [47]:

$$Elongation = 1 - \sum_{i=1}^{NDP} \frac{Length\ of\ minor\ axis_i}{Length\ of\ major\ axis_i} \qquad (3)$$

Additionally, as shown in Equation (3), an average elongation across all precipitation objects within the domain is evaluated. Minimum, maximum and median elongation quantities were also considered but did not provide additional information about the characteristics of the elongation of precipitation objects in the domain.

### 2.3.4. Additional Metrics

Lastly, this analysis includes the number of distinct polygons (NDP) as well as the average area, average perimeter, and average centroid locations of all distinct polygons in the binary precipitation field. The area provides important information about the spatial coverage of each polygon, as well as the overall precipitation coverage when combined with the number of polygons. Perimeter offers insight into the complexity of the polygon shape: a polygon with a larger perimeter will tend to have a less circular shape and/or a more ragged edge. Lastly, the centroid locations are utilized to detect any position biases in the model forecast versus the observed precipitation field.

### 2.4. Precipitation Thresholds

This study aims to better understand how model performance varies by rain rate, particularly when evaluating the representation of extreme rainfall. To gain insight into the HRRR model performance at a range of rain rates, an object-based approach is employed, which requires delineation of a binary precipitation region. For a given one-hour precipitation distribution (Figure 3a), the binary precipitation region is created by retaining only precipitation greater than a defined threshold (Figure 3b). In this study, six precipitation thresholds are utilized: 0.254 mm (0.01 in), 2.54 mm (0.1 in), 6.35 mm (0.25 in), 12.7 mm (0.5 in), 25.4 mm (1 in), and 50.8 mm (2 in) (Figure 4). These thresholds are selected to span the range from "trace" precipitation (0.01 in) to the extreme rain rates observed during the two Ellicott City floods. Additionally, these thresholds cover rain rates that have been traditionally used to evaluate precipitation skill (e.g., [51]). Finally, these precipitation thresholds span both stratiform ($<5$ mm h$^{-1}$) and convective ($>5$ mm h$^{-1}$) rain rates [52,53] and, therefore, offer an opportunity to evaluate whether the HRRR model is more skillful at predicting precipitation corresponding to one of these precipitation regimes. As a final note, it is important to consider that the Stage IV precipitation analysis may not resolve the most extreme precipitation rates due to its one-hour time step and 4 km horizontal resolution. For example, a single cell thunderstorm is approximately 1–10 km in horizontal extent with a lifecycle of 30 min to 1 h. Similarly, individual cells within mesoscale convective systems exhibit comparable spatiotemporal scales [54,55].

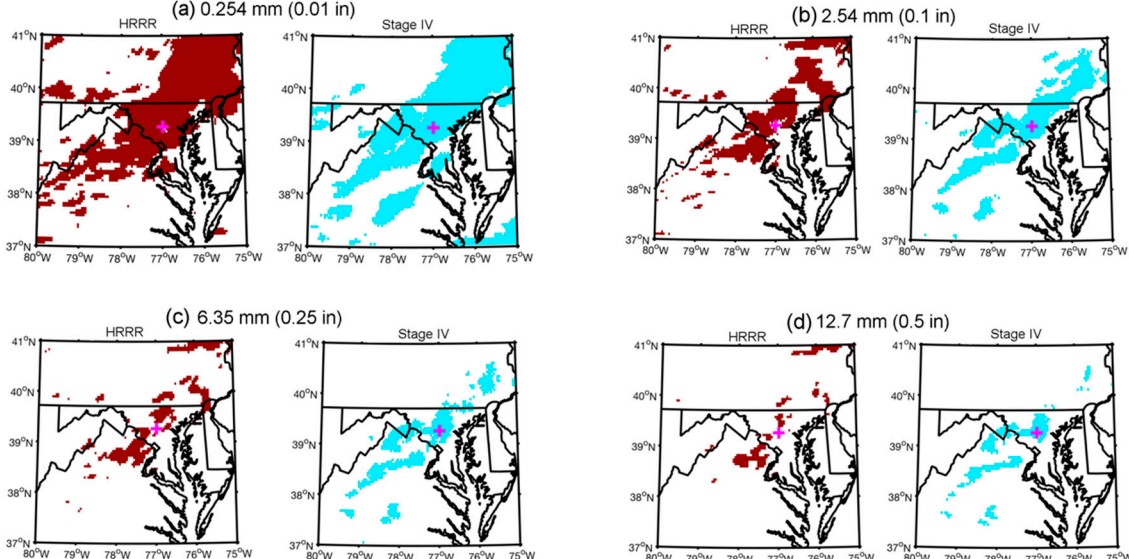

**Figure 4.** Example of delineation of objects at variable thresholds: (**a**) 0.254 mm, (**b**) 2.54 mm, (**c**) 6.35 mm, and (**d**) 12.7 mm. In each panel, the left image corresponds to the HRRR one-hour precipitation forecast, and the right image corresponds to the Stage IV analyzed hourly precipitation.

## 2.5. Statistical Analysis

After measuring the spatial attributes of the precipitation field (as outlined in Section 2.3), the next step is to compare the quantities from the Stage IV analysis with the HRRR model forecast. For each HRRR model cycle, a table is created with the spatial metrics at each hourly forecast interval. Thus, the skill of each HRRR cycle is evaluated for the spatial attributes of precipitation over the forecast period at hourly intervals beginning with the one-hour forecast and ending with time of the extreme rainfall in Ellicott City (ending 23 UTC for 30 July 2016 event and 21 UTC for the 27 May 2018 event). These spatial metrics are then compared with those derived from the Stage IV observations at the corresponding times. Using the 30 July 2016 case as an example, Figure 5 shows the distribution of spatial metrics calculated for the 0.254 mm h$^{-1}$ threshold for the 12 UTC HRRR model cycle versus those same metrics for the Stage IV observations.

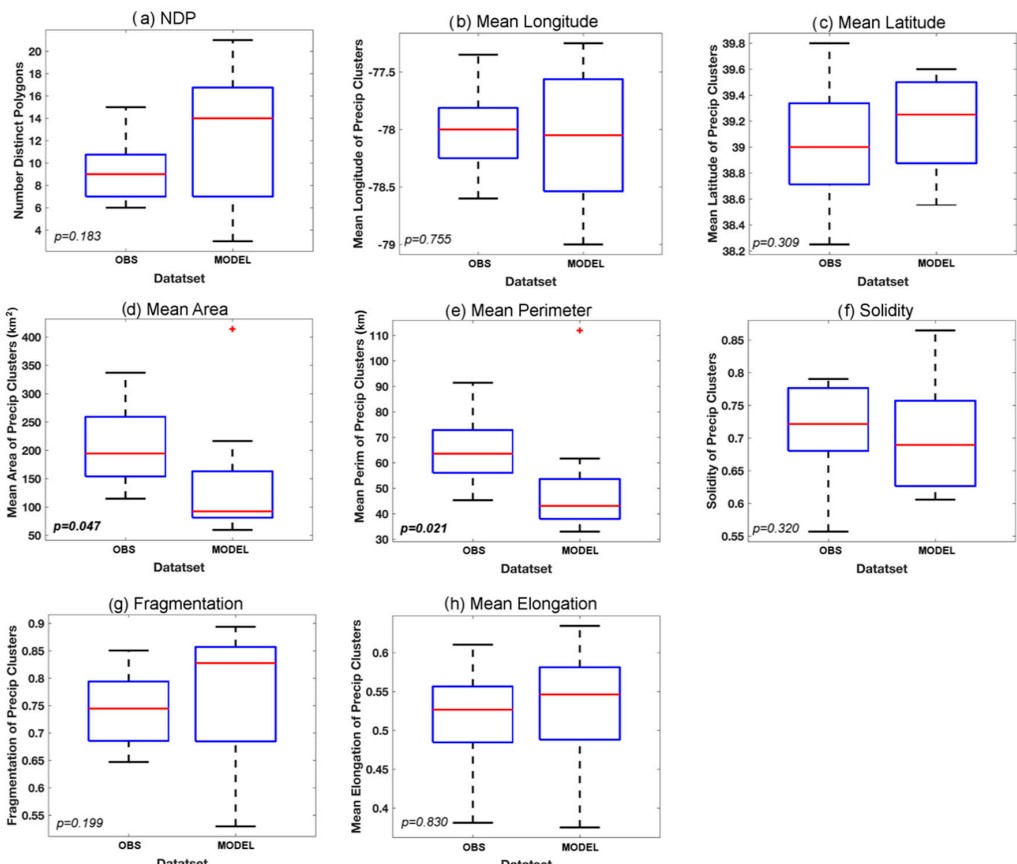

**Figure 5.** Box plots showing distributions of object-based metrics calculated for the 0.254 mm h$^{-1}$ threshold for the 12 UTC HRRR model cycle (MODEL) versus those same metrics for the Stage IV observations (OBS) for the 30 July 2016 event. The corresponding Mann–Whitney U-test *p*-values are included in italicized text in the lower left corner of each plot. For *p* < 0.1, these *p*-values are also in bold face font.

For each model cycle, a Mann–Whitney test is applied. The Mann–Whitney *U* is a nonparametric statistic for comparing the distributions of two unpaired samples via changes in the median [56]. Specifically, the test is applied to ordinal data. The null hypothesis is that there is no difference in the mean ranks of the two samples. When the *U* statistic exceeds a critical value, determined by the *p*-value, the null hypothesis may be rejected in favor of the alternative hypothesis, which states that the differences are too large to be attributed to random sampling [57]. In this study, a 90% confidence level is selected to capture greater degrees of uncertainty than the traditional 95% confidence level, due in part to the strong performance of the HRRR model forecast versus observations. Thus, when *p* < 0.1

for a particular metric, the data indicate that a significant difference exists in the spatial nature of the precipitation distribution corresponding to that metric. Some scientists and statisticians have noted that the selection of a *p*-value for statistical significance is largely arbitrary [58] and often misunderstood [59]. All Mann–Whitney *U*-test *p*-values are reported for reference, leaving space to interpret the results with a stricter threshold, such as $p < 0.05$. In the example provided in Figure 5, there are significant differences in the mean area ($p = 0.047$) and mean perimeter ($p = 0.021$) of precipitation clusters. A post-hoc comparison of these differences indicates that individual rainfall objects in the HRRR are significantly smaller than those in the Stage IV. For this particular model cycle (12 UTC 30 July), there are no other significant differences in the number, location, or spatial distribution of 0.254 mm h$^{-1}$ precipitation clusters.

## 3. Results

### *3.1. Case 1: 30 July 2016*

#### 3.1.1. Traditional Verification Method

For the 30 July 2016 case, in which the precipitation occurred between 22:00 UTC 30 July and 01:00 UTC 30 July HRRR model cycles initialized from 12:00 UTC to 22:00 UTC 30 July (or the 10 h preceding onset of the event) are utilized. First, the two-dimensional (2D) histogram is considered to compare the frequency of observed versus model forecast rain rates (Figure 6a). This precipitation diagnostic is a pixel-based verification method and the results suggest a very poor performance, as revealed by a low frequency of the model forecast and observations in close agreement along the 1:1 line. A closer examination suggests that there may be a location error, since HRRR model forecast and the Stage IV analysis display similar frequencies with respect to each rain rate. For example, approximately 0.25% of pixels include Stage IV rain rates of 4 mm h$^{-1}$ and HRRR model forecast rain rates of 0 mm h$^{-1}$. Yet, a similar but slightly lower frequency exists for HRRR model rates of 4 mm h$^{-1}$ and Stage IV rain rates of 0 mm h$^{-1}$. Thus, the histogram suggests a mismatch in the location of these high rain rate values. Additionally, there is a slight underestimation of precipitation rates in HRRR compared with the Stage IV.

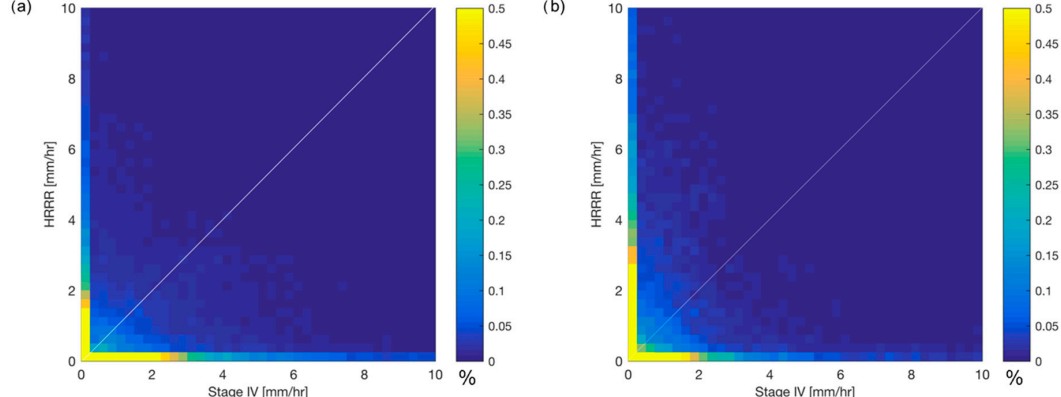

**Figure 6.** Two-dimensional histogram of precipitation from the HRRR hourly precipitation forecast versus Stage IV analyzed hourly precipitation for (**a**) the 30 July 2016 event and (**b**) 27 May 2018 event. The white diagonal line corresponds to a 1:1 line, where the model forecast would verify perfectly with the observation at all locations in the forecast domain.

#### 3.1.2. Object-Based Verification Method

An object-based approach should offer additional insight into the spatial characteristics of the precipitation in the two datasets. A summary of all significant biases for each threshold is provided in Figure 7, with individual threshold results presented in Tables 2–5. First, the representation of trace

rain rates (0.254 mm h$^{-1}$, Table 2) is examined. The Mann–Whitney *U* statistics reveal a significant location bias with respect to the mean centroid latitude and longitude. A post-hoc comparison suggests that the HRRR model generally placed these precipitation regions slightly to the north and east of the observations, particularly in the earlier initialization times. Spatial metrics also reveal numerous biases. Precipitation clusters are consistently too small, too empty, and too fragmented with perimeters that are too smooth. By examining the components of fragmentation, which is the product of solidity and connectivity (Equation (2)), the bias toward fragmented clusters is likely due to the clusters being too empty (e.g., too many holes) and not due to disconnection (e.g., splintering) of clusters.

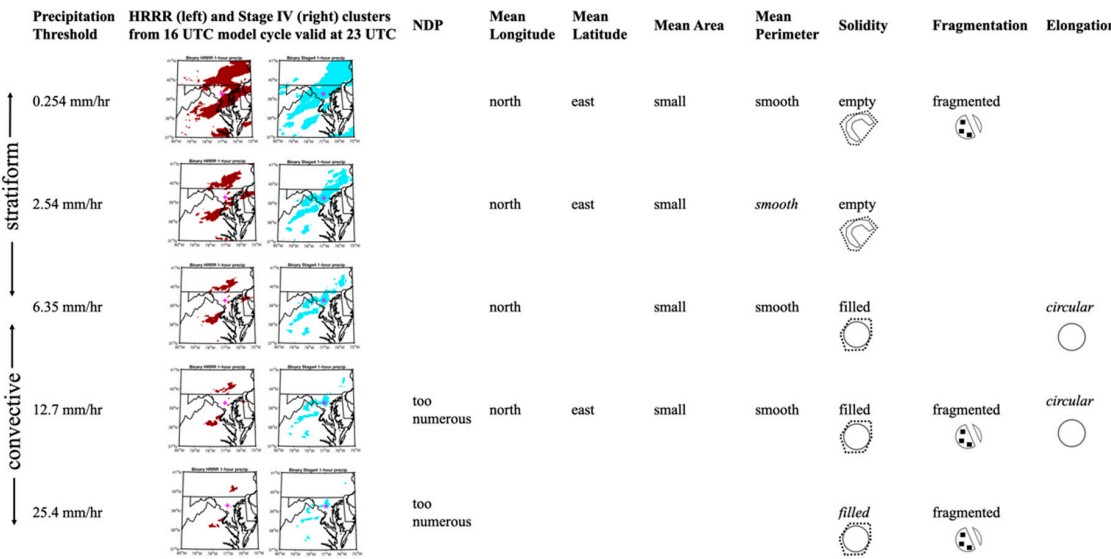

**Figure 7.** Summary of significant (*p* < 0.1) biases in HRRR model forecast relative to Stage IV observations for the 30 July 2016 case study. To illustrate the biases, the 16 UTC model forecast and observations valid at 23 UTC are provided for each precipitation threshold. Consistent biases (identified over multiple model cycles) are in normal font and more intermittent biases (identified in one or two model cycles) are in italicized font.

At the 6.35 mm h$^{-1}$ threshold (Table 3, Figure 7), biases are similar to those at the 0.254 mm h$^{-1}$ threshold but slightly less persistent. The position bias is dominated by latitude, with the HRRR model forecast predicting clusters that are displaced slightly to the north of the observations. At this threshold, there is no consistent longitude bias. Spatially, the precipitation clusters are too small, smooth, and filled in the earlier model cycles. Finally, there is a slight elongation bias in which the HRRR model predicts precipitation clusters that are too circular. This bias is inconsistently analyzed and would not be considered significant at the stricter 95% confidence threshold.

The object-based location and spatial metrics based on 12.7 mm h$^{-1}$ polygons (Table 4, Figure 7) show widespread biases across all model cycles. First, there is a consistent northeast bias in the HRRR forecast precipitation. Additionally, the forecast polygons are too numerous, too small, too solid, and too fragmented. The bias toward solid yet fragmented clusters suggests that the forecast precipitation pattern is too disconnected compared with the observations, which is supported by the bias toward too many polygons. Overall, there is no systematic bias in total area of precipitation at the 12.7 mm h$^{-1}$ threshold (not shown), but individual polygons are too small and too numerous. The perimeters also tend to be a little too smooth. Collectively, the combination of too numerous, smooth polygons with smooth edges suggests that the model forecast precipitation is a little too cellular at this threshold. However, there is not a consistent bias in mean elongation toward circular clusters, which would corroborate this finding.

**Table 2.** Mann–Whitney $U$ $p$-values for 0.254 mm h$^{-1}$ (0.01 in h$^{-1}$) threshold for Jul 31, 2016 case. Bold font face indicates significance at 90% confidence threshold.

| HRRR Initial Time | NDP | Mean Lon | Mean Lat | Mean Area | Mean Perimeter | Solidity | Fragmentation | Elongation |
|---|---|---|---|---|---|---|---|---|
| 12 | 0.183 | 0.755 | 0.309 | **0.047** | **0.021** | 0.320 | 0.199 | 0.830 |
| 13 | 0.405 | **0.069** | **0.043** | **0.026** | **0.059** | 0.161 | **0.070** | 0.696 |
| 14 | 0.156 | **0.085** | **0.005** | **0.019** | **0.025** | **0.036** | **0.024** | 0.442 |
| 15 | 0.728 | **0.003** | **0.001** | **0.019** | 0.429 | **0.017** | **0.011** | 0.977 |
| 16 | 0.291 | **0.003** | **0.087** | 0.117 | 0.793 | **0.076** | **0.025** | **0.017** |
| 17 | 0.400 | **0.037** | **0.075** | 0.212 | 0.307 | **0.031** | **0.097** | 0.970 |
| 18 | 0.919 | **0.008** | **0.081** | **0.069** | 0.863 | **0.024** | **0.002** | 0.730 |
| 19 | 0.553 | 0.978 | 0.720 | 0.382 | 1.000 | **0.021** | 1.000 | 0.279 |
| 20 | 0.693 | 0.329 | 0.878 | 0.259 | 0.318 | **0.017** | **0.046** | 0.710 |
| 21 | 0.182 | 0.351 | 0.576 | 0.132 | **0.041** | **0.002** | **0.009** | 0.589 |
| 22 | 0.333 | 0.690 | 0.548 | **0.031** | **0.008** | **0.008** | **0.032** | 1.000 |

**Table 3.** Mann–Whitney $U$ $p$-values for 6.35 mm h$^{-1}$ (0.25 in h$^{-1}$) threshold for Jul 31, 2016 case. Bold font face indicates significance at 90% confidence threshold.

| HRRR Initial Time | NDP | Mean Lon | Mean Lat | Mean Area | Mean Perimeter | Solidity | Fragmentation | Elongation |
|---|---|---|---|---|---|---|---|---|
| 12 | 0.505 | 0.901 | 0.013 | 0.362 | **0.021** | 0.868 | 0.534 | **0.081** |
| 13 | 0.578 | 0.765 | 0.153 | **0.051** | **0.026** | **0.021** | 0.505 | 0.421 |
| 14 | 0.857 | 0.758 | **0.075** | **0.047** | **0.013** | **0.032** | 0.538 | 0.608 |
| 15 | 0.749 | 0.140 | **0.077** | **0.047** | 0.507 | **0.003** | 0.470 | 0.112 |
| 16 | 0.843 | **0.020** | **0.017** | 0.922 | 0.844 | **0.023** | 0.694 | 0.470 |
| 17 | 0.362 | 0.471 | 0.150 | 0.450 | 0.623 | 1.000 | 0.273 | 0.307 |
| 18 | 0.529 | 0.847 | **0.044** | 0.931 | 0.666 | 0.931 | 0.489 | 0.340 |
| 19 | 0.856 | 0.111 | 0.289 | 0.195 | 0.574 | 0.878 | 0.878 | 0.798 |
| 20 | 0.311 | 0.104 | 0.876 | 0.209 | 0.259 | 0.902 | 0.318 | 0.710 |
| 21 | **0.022** | 0.368 | 0.134 | 0.394 | 0.394 | 0.818 | 0.852 | 0.240 |
| 22 | 0.151 | 0.690 | **0.095** | 0.841 | 0.548 | 0.310 | 0.222 | **0.095** |

**Table 4.** Mann–Whitney $U$ $p$-values for 12.7 mm h$^{-1}$ (0.5 in h$^{-1}$) threshold for Jul 31, 2016 case. Bold font face indicates significance at 90% confidence threshold.

| HRRR Initial Time | NDP | Mean Lon | Mean Lat | Mean Area | Mean Perimeter | Solidity | Fragmentation | Elongation |
|---|---|---|---|---|---|---|---|---|
| 12 | 0.734 | 0.291 | **0.042** | 0.589 | **0.010** | **0.001** | 0.137 | 0.552 |
| 13 | 0.852 | 0.945 | 0.836 | 0.312 | **0.034** | **0.029** | 0.111 | **0.058** |
| 14 | 0.814 | 0.258 | 0.877 | 0.200 | 0.238 | **0.061** | 0.505 | 0.758 |
| 15 | 0.599 | 0.125 | 0.562 | 0.299 | 0.285 | 0.341 | 0.624 | 0.707 |
| 16 | 0.406 | **0.002** | 0.391 | 0.264 | 0.131 | 0.358 | 0.358 | 0.555 |
| 17 | **0.093** | 0.545 | **0.034** | **0.032** | 0.623 | **0.023** | **0.064** | 0.427 |
| 18 | 0.162 | 0.503 | 0.229 | **0.082** | **0.053** | **0.091** | **0.072** | 0.387 |
| 19 | **0.056** | **0.040** | **0.060** | **0.028** | 0.235 | 0.505 | **0.083** | 0.442 |
| 20 | **0.048** | **0.005** | **0.023** | 0.210 | 0.128 | **0.041** | **0.073** | 0.902 |
| 21 | **0.011** | **0.002** | **0.058** | 0.191 | 0.240 | 0.153 | **0.026** | 0.485 |
| 22 | **0.010** | 0.690 | **0.087** | **0.027** | **0.019** | **0.095** | **0.090** | 0.917 |

**Table 5.** Mann–Whitney U Statistics for 25.4 mm h$^{-1}$ (1 in h$^{-1}$) threshold for Jul 31, 2016 case. Bold font face indicates significance at 90% confidence threshold.

| HRRR Initial Time | NDP | Mean Lon | Mean Lat | Mean Area | Mean Perimeter | Solidity | Fragmentation | Elongation |
|---|---|---|---|---|---|---|---|---|
| 12 | 0.569 | 0.152 | 0.223 | 0.798 | 0.328 | 0.130 | 0.161 | 0.574 |
| 13 | 0.599 | 0.486 | 0.671 | 0.877 | 0.847 | 0.877 | 0.699 | 0.298 |
| 14 | **0.019** | 0.468 | 0.919 | 0.252 | **0.056** | 0.142 | **0.013** | 0.482 |
| 15 | **0.012** | 0.656 | 0.708 | 0.151 | 0.104 | 0.613 | **0.015** | 0.860 |
| 16 | **0.031** | 0.219 | 0.867 | 0.302 | 0.161 | 0.490 | **0.027** | 0.962 |
| 17 | 1.000 | 0.710 | 0.364 | 0.453 | 0.805 | 0.209 | 0.456 | 1.000 |
| 18 | 0.448 | 0.974 | 0.413 | 0.853 | 0.485 | 0.485 | 0.310 | 0.699 |
| 19 | 0.285 | 0.551 | 0.197 | 0.639 | 0.432 | 0.755 | 0.343 | 0.876 |
| 20 | **0.070** | 0.654 | 0.604 | 0.563 | 0.200 | 0.432 | 0.202 | 0.530 |
| 21 | **0.100** | 0.331 | 0.887 | 0.329 | 0.329 | 0.662 | 0.102 | 0.931 |
| 22 | 0.405 | 0.114 | 0.429 | 0.943 | 0.450 | 0.686 | 0.623 | 0.886 |

Due to the small number of precipitation clusters at higher rain rates, the Mann–Whitney U-test could not be applied at the highest rain rates. Therefore, the 25.4 mm h$^{-1}$ threshold must be used to assess the spatial characteristics of the most extreme rain rate regions. Interestingly, the HRRR model performed very well at this rain rate threshold, with only a few significant biases. The most notable bias is a consistent tendency toward predicting too many precipitation clusters. Additionally, the overall pattern is too fragmented. These two results are similar to some of the spatial characteristics at the 12.7 mm h$^{-1}$ rain rate threshold, suggesting that there may be some continuity in these higher rain rate regions. Interestingly, the results also show a transition from a bias toward empty clusters for stratiform rain rates to a bias toward filled clusters for convective rain rates (Figure 7). There is no location bias for the 25.4 mm h$^{-1}$ threshold, which may indicate that the location bias is confined to the lower rain rate thresholds.

3.1.3. Forecasting Context for Case 1

Results for the 30 July 2016 case study are summarized in Figure 8 with a snapshot of the Stage IV one-hour precipitation (Figure 8a) and HRRR model forecast one-hour precipitation (Figure 8b) totals, both valid for the period ending 23 UTC 30 July or in the middle of the extreme precipitation event. With this visual verification, it is possible to detect some of the biases that were revealed by the object-based precipitation verification results. For instance, the northeast bias in location for the lower rain rate regions is evident by the HRRR rainfall forecast concentrated more over southeastern Pennsylvania compared with northern Virginia. Additionally, the model forecast appears to include more isolated cells compared with the Stage IV precipitation clusters.

To place these results in context, an overview of the synoptic environment is provided at the same valid time, as forecast by the 12 UTC HRRR model cycle (Figure 8c) and observed by the WPC in its mesoscale precipitation discussion (Figure 8d). The 12Z HRRR model cycle is selected for its relatively strong performance versus other model cycles (Tables 2–5). This strong performance may be due in part to a its skillful representation of the large-scale environment (Figure 8c,d). In particular, the 12 UTC cycle and later model cycles provide the best representations of the weak low pressure over southern Michigan and the weak high ridge off the New England coast (Figures 1b and 8c,d). Therefore, the region of convergence at approximately 23 UTC Jul 30 was fairly well represented, which was crucial to providing moisture to support of the back-building line convective cells that impacted the Ellicott City area. Other model cycles, particularly the early model cycles, were not as consistent in the representation of this large-scale set-up, and therefore, the model forecast precipitation sometimes suffered as a result.

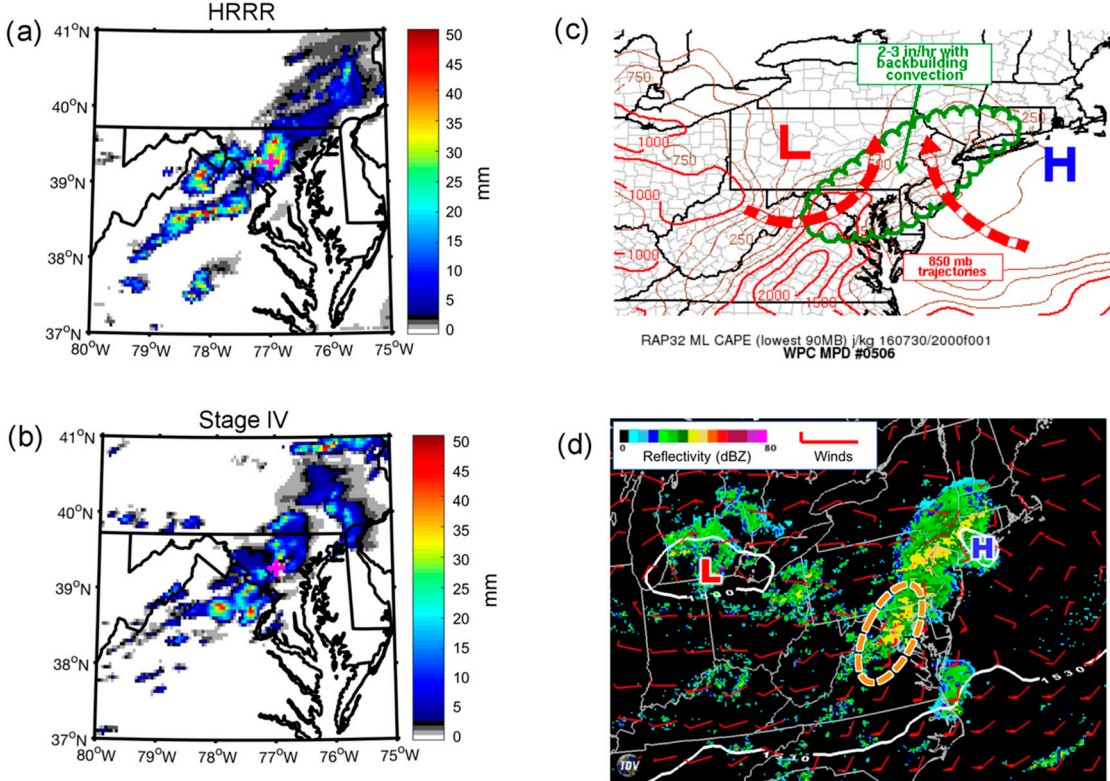

**Figure 8.** Summary for 30 July 2016: (**a**) Stage IV one-hour precipitation totals valid for the period ending 23 UTC 30 July (**b**) 12 UTC HRRR one-hour precipitation forecast valid for the period ending 23 UTC 30 July (**c**) WPC analysis associated with a mesoscale precipitation discussion issued at 17:00 EDT on 30 July 2016, and (**d**) 12 UTC HRRR forecast 850 mb winds (red vectors), geopotential heights (white contours), and model simulated composite reflectivity valid at 23 UTC Jul 30 (image made in IDV). In (**d**) a region of convergence over central Maryland is indicated by an orange dashed oval.

*3.2. Case 2: 27 May 2018*

3.2.1. Traditional Precipitation Verification

In contrast with the 2016 event, the 27 May 2018 precipitation occurred in two waves, with the first wave beginning around 1900 UTC. As with the prior case, the HRRR model forecasts cycles are examined beginning 10 h before onset of the heaviest precipitation. Therefore, this analysis includes model cycles initialized between 09 UTC and 19 UTC 27 May. The 2D histogram for this event (Figure 6b) again suggests a poor performance when comparing the observed and model forecast rain rates. Similar to the 2016 case, there is a mismatch in location of the heaviest rainfall, which leads to poor skill as represented by this pixel-based verification method. However, in this second case, the 2D histogram indicates an additional issue with the forecast, since a majority of the data are located to the left of the 1:1 line, suggesting that the HRRR model overestimated the precipitation rates across the region in this event.

3.2.2. Object-Based Precipitation Verification by Threshold

For this second case study, the Mann–Whitney U statistics are presented for two rain rate thresholds, since the lower threshold results are highly correlated with one another (not shown). As with the 2016 case study, a summary of significant results is also presented in Figure 9. First, it is important to note that there is no consistent location bias at any rain rate threshold (Tables 6 and 7). At the lowest rain rates (Table 6, Figure 9), the earlier model cycles consistently overestimate the number of precipitation polygons and the pattern is generally too fragmented. There is also a slight bias toward polygons that

are too small, too ragged, and too empty. At higher rain rates (Table 7, Figure 9), the earlier model cycles performed slightly better than later model cycles. In these later model cycles, the HRRR has a tendency toward producing too many precipitation polygons that are too empty and fragmented. Some of the model cycles also produce polygons that are too large with edges that are too ragged. Unlike with the first case study, the solidity bias does not reveal differences for stratiform vs. convective rain rates. Instead, the clusters tend to be too empty across all rain rate thresholds. Unfortunately, there are too few Stage IV observations of rain rates greater than 25.4 mm h$^{-1}$ (1 in h$^{-1}$), preventing a more in-depth comparison of the spatial characteristics of the heaviest rain rate regions.

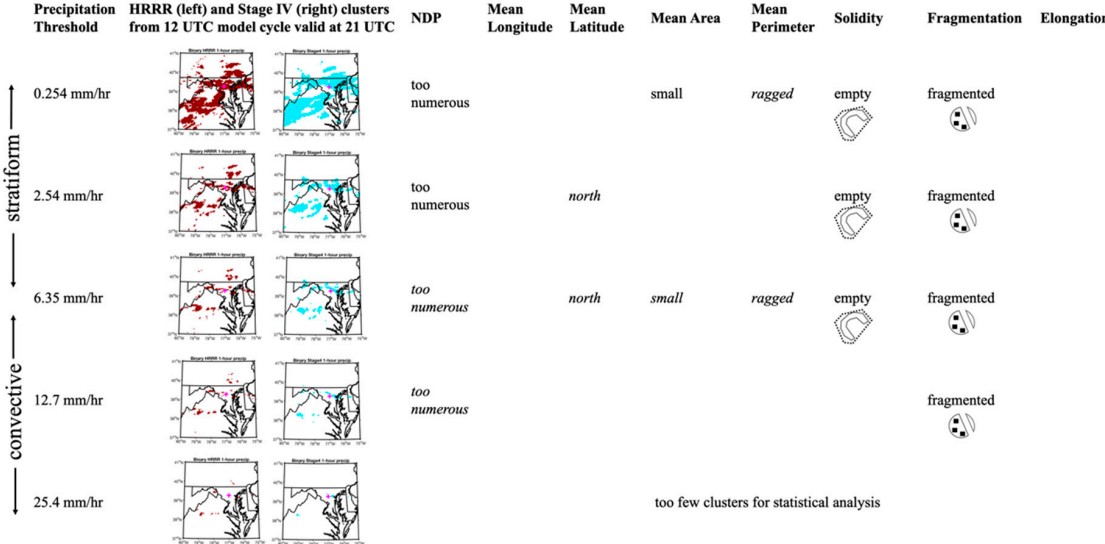

**Figure 9.** Summary of significant ($p < 0.1$) biases in HRRR model forecast relative to Stage IV observations for the 27 May 2018 case study. To illustrate the biases, the 12 UTC model forecast and observations valid at 21 UTC are provided for each precipitation threshold. Consistent biases (identified over multiple model cycles) are in normal font and more intermittent biases (identified in one or two model cycles) are in italicized font.

**Table 6.** Mann–Whitney *U* *p*-values for 0.254 mm h$^{-1}$ (0.01 in h$^{-1}$) threshold for 27 May 2018 case. Bold font face indicates significance at 90% confidence threshold.

| HRRR Initial Time | NDP | Mean Lon | Mean Lat | Mean Area | Mean Perimeter | Solidity | Fragmentation | Elongation |
|---|---|---|---|---|---|---|---|---|
| 9 | **0.004** | 0.783 | 0.370 | 0.448 | 0.730 | 0.223 | **0.002** | 0.696 |
| 10 | **0.018** | 0.174 | 0.105 | 0.218 | 0.837 | **0.024** | **0.003** | 0.124 |
| 11 | **0.002** | 0.225 | 0.795 | **0.019** | 0.141 | 0.795 | **0.004** | 0.371 |
| 12 | **0.040** | 0.263 | 0.278 | **0.057** | **0.076** | 0.646 | **0.026** | 0.743 |
| 13 | **0.093** | 0.733 | 0.363 | 0.121 | 0.212 | 0.521 | **0.076** | 0.427 |
| 14 | **0.008** | 0.197 | 0.814 | 0.258 | 0.489 | 0.730 | **0.001** | 0.489 |
| 15 | 0.149 | 0.110 | 0.627 | 0.959 | 0.878 | 0.328 | **0.038** | 0.645 |
| 16 | 0.196 | 0.362 | 0.710 | 0.902 | 1.000 | 0.456 | 0.259 | 0.902 |
| 17 | 0.807 | 0.604 | 0.680 | 0.937 | 0.589 | 0.132 | 0.699 | 0.180 |
| 18 | 1.000 | 0.103 | 0.571 | 0.310 | 0.421 | 0.310 | 0.841 | 0.690 |
| 19 | 0.886 | 0.200 | 0.114 | 1.000 | 0.886 | 0.686 | 0.686 | 0.486 |

**Table 7.** Mann–Whitney $U$ $p$-values for 12.7 mm h$^{-1}$ (0.5 in h$^{-1}$) threshold for 27 May 2018 case. Bold font face indicates significance at 90% confidence threshold.

| HRRR Initial Time | NDP | Mean Lon | Mean Lat | Mean Area | Mean Perimeter | Solidity | Fragmentation | Elongation |
|---|---|---|---|---|---|---|---|---|
| 9 | 0.744 | 0.251 | **0.011** | 0.326 | 0.211 | 0.345 | 0.661 | 0.968 |
| 10 | 0.400 | 0.131 | 0.309 | 0.369 | 0.277 | 0.655 | 0.541 | 0.911 |
| 11 | 0.766 | 0.896 | 0.584 | 0.999 | 0.681 | 0.391 | 0.918 | 0.681 |
| 12 | 0.676 | 1.000 | **0.094** | 0.366 | 0.836 | 0.347 | 0.731 | 0.534 |
| 13 | 0.476 | 0.910 | 0.971 | 0.945 | 0.731 | 0.418 | 0.366 | 0.295 |
| 14 | 0.948 | 0.865 | 0.122 | 0.890 | 0.689 | 0.310 | 0.955 | 0.607 |
| 15 | 0.462 | 0.294 | 0.729 | 0.300 | 0.345 | 0.325 | 0.282 | 0.950 |
| 16 | 0.337 | 0.699 | 0.704 | 0.138 | **0.035** | **0.034** | **0.035** | 0.836 |
| 17 | **0.054** | 0.935 | 0.697 | 0.240 | 0.132 | **0.089** | **0.026** | 0.818 |
| 18 | **0.056** | 0.500 | 0.730 | **0.095** | **0.095** | 0.087 | **0.032** | 0.841 |
| 19 | **0.057** | 0.114 | 0.200 | 0.200 | 0.200 | 0.114 | **0.029** | 1.000 |

### 3.2.3. Forecasting Context for Case 2

Visual comparisons of Stage IV one-hour precipitation (Figure 10a) and 15 UTC HRRR model forecast one-hour precipitation (Figure 10b), both valid at 23 UTC on 30 July show some of the biases that were detected with the object-based analysis. First, there appear to be too many precipitation clusters, including small clusters of light rain across southwestern portions of the domain (also evident in Figure 9), which contributed to the higher fragmentation. Second, the HRRR forecast includes too many regions with higher rain rates, particularly in areas outside of Ellicott City. The HRRR forecast bias toward overestimating precipitation within these higher rain rate regions is also evident. It should be noted, however, that the Stage IV analysis may have struggled to capture the extreme rain rates that were recorded at isolated locations within the domain, particularly in the Ellicott City area. Lastly, unlike the 2016 case, there does not appear to be a location bias.

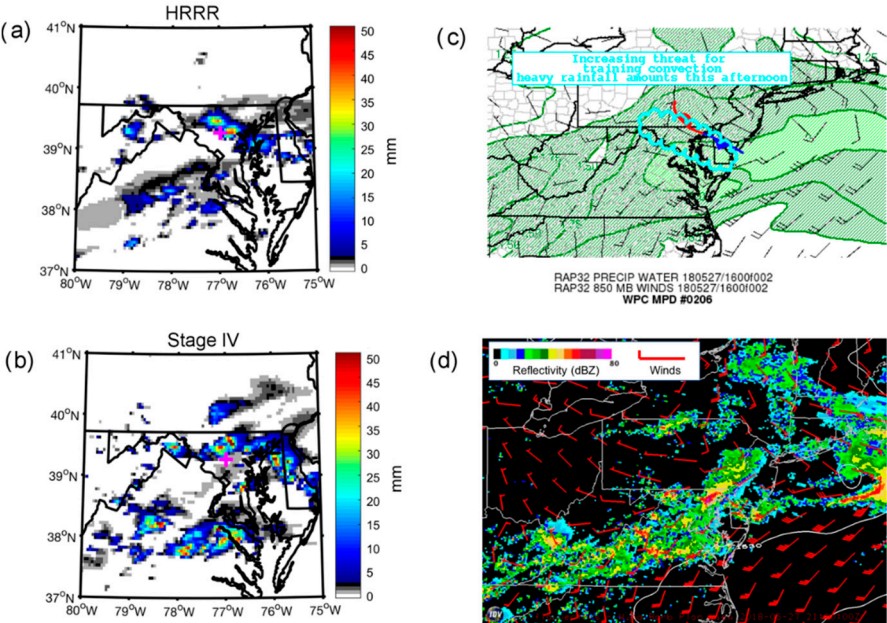

**Figure 10.** Summary for 27 May 2018: (**a**) Stage IV one-hour precipitation totals valid for the period ending 21 UTC 27 May (**b**) 15 UTC HRRR one-hour precipitation forecast valid for the period ending 21 UTC 27 May (**c**), WPC analysis associated with a mesoscale precipitation discussion issued at 13:36 EDT on 27 May 2018, and (**d**) 15 UTC HRRR forecast 850 mb winds (red vectors), geopotential heights (white contours), and model simulated composite reflectivity valid at 21 UTC May 27 (image made in IDV).

Examining the model's representation of synoptic-scale features (Figure 10c,d), Subtropical Storm Alberto is a prominent feature, as well as the broad region of southeasterly flow. Additionally, the frontal boundary situated to the north of Ellicott City along the Pennsylvania-Maryland border is evident based on the wind shift from southeasterly (to the south of the boundary) to northwesterly (to the north of the boundary). All HRRR model cycles were fairly consistent in the representation of these features, which likely provided confidence in the evolution of these features as meteorologists issued watching and warnings about the flash flood risk in the Ellicott City area. Thus, the HRRR model provided a skillful forecast regarding the synoptic environment that produced heavy rains in Maryland on this day. At the same time, it is important to note a slight bias toward producing rain rates in excess of those that were observed.

### 3.3. Influence of Precipitation Threshold

In this study, the second research question focused on evaluating HRRR model forecast skill over a range of rain rate thresholds. An underlying hypothesis was that the skill would vary by threshold, especially when comparing results for stratiform vs. convective precipitation. Figures 7 and 9 provide some insight into evaluating this hypothesis. Indeed, as noted in the 2016 case study, there was a tendency for the HRRR model to predict precipitation clusters that were too empty for stratiform rain and too filled for convective rain. When considering the skill at each threshold, it is important to note that the HRRR model underwent a significant upgrade (Table 1), which makes it difficult to draw broad conclusions. Still, there are a few consistent biases across both case studies. For example, the precipitation field tends to be too fragmented across all thresholds, with too numerous clusters that are too small compared with the observations. Otherwise, there are some inconsistencies between the two studies, including a tendency toward perimeters that are too smooth in the 2016 flood event and too ragged in the 2018 flood event, regardless of precipitation threshold. Lastly, at the higher rain rate thresholds, there are discrepancies in the solidity bias in 2016 vs. 2018.

To further evaluate the role of precipitation threshold, Spearman's rank correlation coefficients are calculated to compare the correspondence between each object-based metric in the Stage IV and HRRR datasets (Figure 11). Spearman's rank correlations are selected due to nonlinear relations between variables. A stronger positive correlation indicates a monotonic relationship between variables. These correlations reveal a few major findings that are consistent with the Mann–Whitney U-test results: (1) there are stronger correlations for location (mean latitude and mean longitude) in the 2018 case study, (2) the strongest correlations are seen with the elongation metric, and (3) results vary by precipitation threshold and by spatial metric.

There are also a few surprising results. Most notably, correlations tend of be lower at higher rain rate thresholds, particularly for NDP, mean area, solidity, and fragmentation. This result suggests that median values are similar (based on Mann–Whitney $U$-test results) but that the metrics do not covary as strongly at these higher rain rates compared with lower rain rates. In other words, there is not a bias for that metric, yet there is a weaker relationship between the model forecast and the observations. Another somewhat surprising result is the tendency for lower correlations at the lower rain rate threshold ($0.254$ mm h$^{-1}$), a result that is seen in both case studies. Lastly, while many of the correlations improved or remained consistent for the 2018 event, the elongation correlations decreased slightly, especially at the higher rain rate thresholds. This is a surprising result, given the model upgrades, but many external factors could contribute to this result.

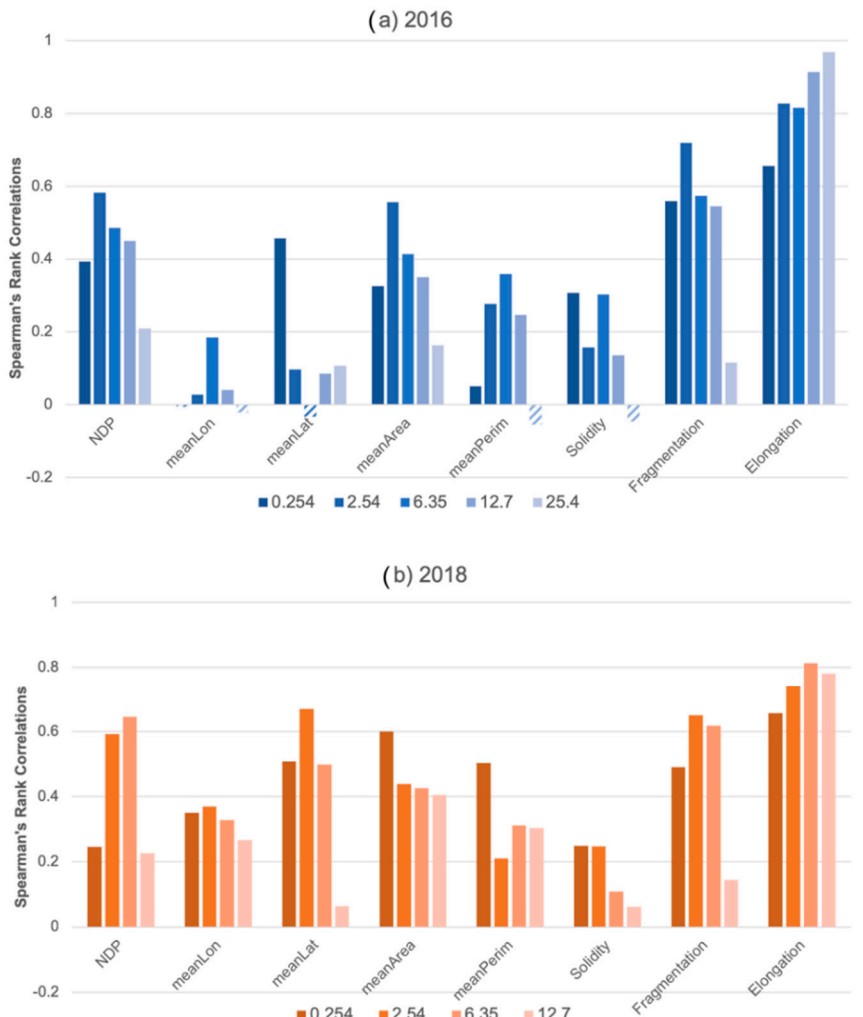

**Figure 11.** Spearman's rank correlation coefficients between HRRR and Stage IV object-based metrics subset by precipitation threshold for all model cycles for the (**a**) 2016 and (**b**) 2018 case study. Correlations that do not meet a 95% confidence threshold are denoted by hatched filling.

## 4. Discussion

The historic rain rates associated with the Ellicott City floods presented a unique opportunity to examine high-resolution NWP skill in predicting extreme rainfall rates. As noted in the introduction, it is expected that anthropogenic climate change will lead to more frequent extreme rainfall events, similar to those evaluated in this study. Because these extreme events severely tax the urban hydraulic infrastructure, it is crucial to assess model skill in characterizing the precipitation forecast. In particular, end users will be interested in understanding model skill at higher rain rate thresholds.

These two case studies highlight that a skillful precipitation forecast requires a skillful representation of synoptic-scale and mesoscale atmospheric processes, such as the large-scale moisture and wind fields and the locations of frontal boundaries. For both case studies, the HRRR model generally provided a good representation of the large-scale environment, and therefore, object-based metrics revealed that there was generally good agreement between model forecast precipitation and observations. However, there important subtle differences between the model and observations in the 2016 event. These subtle differences stress the importance of synoptic-scale and mesoscale processes for both cases. For example, in the first case study, slight differences in the representation of a ridge off the New England coast and the low pressure center over southern Michigan, when compared with the analysis, led to a slight northward displacement of a stationary frontal boundary and, therefore,

a displacement of model forecast precipitation to the north of observed precipitation. In the second case study, where synoptic- and mesoscale features were in close agreement between the model and observations, there was no consistent location bias.

Lastly, the two case studies offer some encouraging results for precipitation forecasts at higher rain rate thresholds, in that the models are generally capable of producing rainfall fields that are consistent with observations at these high rain rates. Yet, there is still room for improvement, since model forecasts of extreme convective rainfall tend to be slightly too numerous and fragmented compared with observations. Of note, in the second case study, the highest rainfall rates may have been poorly resolved by the Stage IV observations, which prevented a more in-depth analysis. This highlights the need for improved observational datasets for verification purposes.

Overall, this study provides evidence that should lend confidence to the quality of high-resolution models and the operational forecast products provided to end users. More specifically, this study indicates that high-resolution models, such as the HRRR, can skillfully predict the extreme precipitation rates that are anticipated with anthropogenic climate change. Still, a more comprehensive analysis of systematic biases over a larger sample of extreme precipitation events is necessary to more carefully evaluate HRRR model forecast precipitation. With a better understanding of high-resolution model skill in predicting extreme precipitation events, meteorologists and emergency managers can work together more effectively to save lives when flash floods threaten communities.

## 5. Conclusions

This study used an object-based method to evaluate hourly precipitation forecasts from the HRRR model against Stage IV observations during two high-impact flash flooding events in Ellicott City, MD. Spatial metrics were utilized to quantify the spatial attributes of precipitation features. A Mann–Whitney U-test then revealed systematic biases (using a confidence level of 90%) related to the shape and location of model forecast precipitation. Additionally, model forecast skill was evaluated over a range of rain rates that span stratiform and convective precipitation regimes.

Results indicated that traditional pixel-based precipitation verification metrics, such as those based on two-dimensional histograms, are limited in their ability to quantify and characterize model skill, due in large part to location differences in the location of precipitation. An object-based approach provided additional useful information model biases with respect to location and spatial structure of the forecast precipitation field.

An important objective of this research was to evaluate model skill over multiple rain rate thresholds. In this limited study based on two cases, the results indicated that model skill was sensitive to rain rate threshold. Yet, the results were also sensitive to the case under consideration, which led to some inconsistencies in the results from the two studies that may indicate that the biases that were observed in one case study may not be systematic to the model. For instance, a location error was only observed in the July 2016 flood, with a slight bias for the model to forecast precipitation at the lower rain rate thresholds to the northeast of the observed locations. No location bias was present at higher rain rate thresholds or for the May 2018 flood. A more consistent issue across both case studies was a tendency for the model to predict a precipitation field that was too fragmented compared with the observations. This bias toward a fragmented precipitation field was present across all thresholds, but most particularly at low rain rate ($<5$ mm h$^{-1}$) and high rain rate ($>10$ mm h$^{-1}$) thresholds. Lastly, the model tended to over-forecast the number of polygons associated with high rain rate thresholds, which contributed to the high fragmentation bias. Interestingly, moderate rain rate (5–10 mm h$^{-1}$) regions in the 2016 flood were one major exception to an otherwise consistent high fragmentation bias: at the 6.35 mm h$^{-1}$ threshold, the polygons tends to be too cellular (solid with smooth edges).

Overall, the results indicate that the HRRR forecast showed some skill in predicting a potential for extreme rainfall in both of the Ellicott City floods, though the model tended to overpredict the rainfall rates in the 2018 flood versus observations. Here, it is important to note that the 4 km Stage IV dataset may have struggled with resolving the high rain rates that were observed over a small area in this

more isolated precipitation event. Still, the object-based approach revealed a tendency to overpredict the number of high rain rate regions.

Both case studies stressed that an accurate forecast of the synoptic and mesoscale environment was crucial to precipitation forecast skill across all rain rate thresholds. For example, the HRRR model struggled more with the synoptic-scale environment in the 2016 case, where a weak high pressure was present to the northeast of the study area. With a region of low pressure over southern Michigan and that high pressure off the New England coast, there was a region of convergence over central Maryland, which promoted the development of backbuilding convection over the Ellicott City area. In the 12 UTC model cycle and in model cycles closely preceding the event, these features were forecast with better skill, and the resulting forecast of convective-scale elements also verified better with the observations. Furthermore, the northeast bias in the location of convection was also related to the representation of these synoptic- and mesoscale weather features. In the 2018 case study, the southerly flow, ample moisture transport, and location of stationary front were more consistently forecast across all model cycles, leading to slightly better skill with respect to the object-based metrics, including a better forecast with respect to the location of precipitation. Overall, the object-based metrics offer evidence that the models are capable of producing extreme precipitation but also that the location and characteristics of that precipitation depend on accurate depiction of synoptic- and mesoscale weather features. These results support that a detailed surface analyses [60] and an "ingredients-based" approach [61] should remain central to the process of forecasting excessive rainfall.

This paper presents two case studies of extreme precipitation events that led to flash flooding in Ellicott City, MD, and therefore there are limitations that arise due to the selection of the cases and the small sample size. A more comprehensive study needs to be performed to evaluate high-resolution precipitation forecasts for such extreme precipitation events. Additionally, future studies need to evaluate precipitation forecast skill over a comprehensive set of synoptic and mesoscale environments to better understand how skill varies by large-scale forcing and storm mode (e.g., linear frontal systems versus isolated cellular storms). Additionally, this study analyzed the model skill over the evolution of the event. An alternate approach would be to utilize a time-lagged ensemble or the new experimental HRRR ensemble to evaluate model skill at individual forecast (valid) times. Lastly, the spatial metrics utilized in this study do not necessarily represent a comprehensive set of independent spatial metrics. More work needs to be conducted to identify the most relevant metrics that capture the diverse range of two-dimensional rainfall patterns associated with mesoscale convection. Thus, this study represents a stepping stone toward a comprehensive evaluation of precipitation in high-resolution forecast models.

**Funding:** This research received no external funding.

**Acknowledgments:** The author is grateful to Steven Quiring for comments on an earlier version of the manuscript, including suggestions to improve the methods section and the accompanying figures. Additionally, two anonymous reviewers provided helpful suggestions, which helped to improve the quality of this manuscript.

**Conflicts of Interest:** The authors declare no conflict of interest.

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
