# Peer review of "Quantifying Extreme Precipitation Forecasting Skill in High-Resolution Models Using Spatial Patterns: A Case Study of the 2016 and 2018 Ellicott City Floods"

_atmosphere, doi:10.3390/atmos11020136_

Round 1

Reviewer 1 Report

The study is very interesting and up to date considering problems of extreme precipitation forecasting. The manuscript is well structured, very good presented and sentences clearly expressed.

I can recommend for publication after minor revision.

Line 6: Affiliation is missing. The main focus in introduction is on increase in frequency of floods associated with urbanization and climate change, and the authors state that both cases provide stark examples for this. How exactly the author links these flood events with urbanization and climate change? Fig.1 – better to draw the area on the synoptic maps corresponding to the upper maps. Line 154: “…synoptic pattern led to south-southwesterly low-level flow over the mid-Atlantic region…” – it looks mainly south-southeasterly wind in the area along the Pennsylvania-Maryland border Description of synoptic condition development is very detailed but not obvious only from the synoptic maps. Maybe radar data can prove the description better. Fig. 5: Keep the numbering of both axes the same, for example only even numbers, like on the X-axis. Lines 221-227: HRRRv1 was operational in July 2016 and HRRRv2 was operational in May 2018. Important upgrades associated with version 2 are summarized in Table 1 and include transition to a newer version of WRF-ARW; significant updates to microphysics, planetary boundary layer, and land surface model parameterization schemes; and new assimilation of radial wind, surface mesonet, and satellite radiance data.      - More of the object-based metrics show improvement (the correlation increase) significantly for more threshold for the mean values, only for the elongation metric in opposite, decreased (Fig.10). Most likely the overall improvement is result from the model upgrades. Any comments why the elongation metric is worse?

Author Response

Thank you for your comments and suggestions. I have considered each comment carefully, and I have added my responses in italics below.

Line 6: Affiliation is missing.

Thank you for catching this. I have added my affiliation.

The main focus in introduction is on increase in frequency of floods associated with urbanization and climate change, and the authors state that both cases provide stark examples for this. How exactly the author links these flood events with urbanization and climate change?

Thank you for this question. I’ve inserted phrase in the introduction to clarify that I am not linking the floods to climate change or urbanization. Rather, I am pointing to these events as the type of events that will be expected in the future.

Fig.1 – better to draw the area on the synoptic maps corresponding to the upper maps.

Thank you for this suggestion. I tried to add a box on the larger synoptic map, but it obscured important text and symbology. Therefore, I have decided to leave the figure as is.

Line 154: “…synoptic pattern led to south-southwesterly low-level flow over the mid-Atlantic region…” – it looks mainly south-southeasterly wind in the area along the Pennsylvania-Maryland border

Description of synoptic condition development is very detailed but not obvious only from the synoptic maps. Maybe radar data can prove the description better.

To address the previous two comments, I have expanded Figure 1 into two separate figures so that the synoptic map could be enlarged and the text and symbology would be easier to read. I explored other options for this synoptic overview, but I had difficulty finding a map or figure that provided a good overview in high enough resolution. I hope that this addresses your concerns and that you can see the south and southwest winds across the study area. (Note that there is a SSE wind barb as well but they are mostly S and SW).

Fig. 5: Keep the numbering of both axes the same, for example only even numbers, like on the X-axis.

Done

Lines 221-227: HRRRv1 was operational in July 2016 and HRRRv2 was operational in May 2018. Important upgrades associated with version 2 are summarized in Table 1 and include transition to a newer version of WRF-ARW; significant updates to microphysics, planetary boundary layer, and land surface model parameterization schemes; and new assimilation of radial wind, surface mesonet, and satellite radiance data.  - More of the object-based metrics show improvement (the correlation increase) significantly for more threshold for the mean values, only for the elongation metric in opposite, decreased (Fig.10). Most likely the overall improvement is result from the model upgrades. Any comments why the elongation metric is worse?

This is an interesting point and something that I hadn’t given much thought to. It might have something to do with the nature of the precip on the two days. One thing to note is that the precipitation was much more isolated for the 2018 event, and perhaps it was a little harder to predict. I have added two sentences in section 3.3 to include this result and a short discussion.

Reviewer 2 Report

Please see attached PDF. 

Author Response

Thank you for your comments and suggestions. I have considered each comment carefully, and I have added my responses in italics below.

Synopsis:

This paper describes an object-based verification approach to evaluate short-range hourly forecast precipitation from the HRRR model versus Stage-IV QPE.

I found this to be a well-written, careful study prototyping an important method (spatially-focused, object-based verification) with which to evaluate two high-impact flash flood events. I think it will be ready for publication pending consideration of a few minor comments below.

General comments:

The approach/software used here seems very similar to the capabilities offered by NCAR/DTC’s MODE tool (which is referenced in the manuscript). Are there specific reasons why MODE wasn’t used here? It seems that there would be value in using a community tool if a goal here is partially to prototype the approach to ultimately engender community support for this type of evaluation.

Thank you for this suggestion. I have been using my own metrics because it gives me some autonomy (and flexibility) in how the metrics are calculated and which metrics are included. In the future, it would be a great idea to use the MODE tool so that other users can more easily follow the work and implement this community tool as a diagnostic for rainfall forecasts.

As no studies have actually shown the Ellicott City floods to have a connection with anthropogenic climate change, I suggest being cautious to not implicitly (over)state such a connection here. Granted, the events are certainly consistent with ACC and in most places this is appropriately couched, but the mention in the abstract took me a bit by surprise (line 23).

This is an excellent point. I have refined the wording so that it no longer implies a direct link with ACC. Instead, I meant for these case studies to represent the types of floods that might be expected with climate change.

Urbanization is mentioned several times (e.g., line 126) as providing ideal conditions for the EC flash flooding. Suggest being more specific about whether this is intended to mean the urban environment in Historic Ellicott City proper, or rather over upstream areas (as development in these areas has been blamed – though not proven – to have been a factor.) It’s a mostly tangential point, but I mention it as a precaution so as to not unintentionally bolster potentially unsubstantiated claims either way.

I’ve removed the direct tie here as well, as this was not my intention.

The notion of these results not precluding the need for detailed surface analysis and an ingredients-based forecast approach is not one that I think most people would quibble with, but I don’t necessarily know that “these results suggest” this directly. (e.g., lines 678 – 680). Is there a way to more strongly connect this study’s results with this idea?

Thank you, I agree that the wording needed to be improved here. I have added another sentence to link that sentence with the results presented in the paper.

Certain key figures could be improved by simply zooming in and enlarging. Given the manuscript’s detail offered on the spatial verification results, I think most readers might desire a better visual, i.e., being able to better see the thumbnail examples in Figs. 6 and 8. Additional other specific figure suggestions below.

Thank you for these suggestions. I have expanded Figure 1 into two separate figures so that the synoptic map could be enlarged and the text and symbology would be easier to read. I have also enlarged Figures 6 and 8 so that the thumbnails should be easier to see. I hope that this change to Figures 6/8 makes it easier for readers to visualize the spatial patterns that are described by the metrics.

Specific suggestions:

Lines 71 – 72: Suggest changing that CAMs including HRRR “may be able to” to “…were developed to capture…”

Done

Figure 1: Can better surface analyses be obtained, or can these at least be zoomed in? You really don’t need all of CONUS shown.

I explored other options for this synoptic overview, but I had difficulty finding a map or figure that provided a good overview in high enough resolution. I did not want to zoom in on the region itself and lose the broad synoptic picture which can also be important. I hope that the larger image makes it easier to see the surface analysis.

Line 197: capitalize “city”

Done

Line 209: “resolution” should be “grid spacing” (e.g., https://journals.ametsoc.org/doi/pdf/10.1175/1520-0477%282000%29081%3C0579%3ACAA%3E2.3.CO%3B2)

Done

Line 236: Add NCAR before Earth Observing Lab

Done

Line 296: “Next” seems a bit strange/awkward – next following what?

Modified to more clearly outline the sequence of events for the methods (After the binary image is delineated…)

Section 3.1.1/general: how many pixels are in the domain?

It is a good idea to include some details about the grid, which is 99 x 81 (8019 pixels). I have added a sentence to section 2.2.

Figure 7: Zooming and enlarging recommended in general

I decided to enlarge some of the other figures but I could not think of a good way to enlarge this one without splitting it into multiple figures. That said, the image is high resolution and I think that the final digital image will be good enough quality that the readers can zoom in if necessary.

Line 545: I believe that there were some notable location errors for some HRRR cycles in the 2018 case, but that they jumped around from cycle to cycle and so they may have been smoothed out if looking for a bias in the aggregate.

Interesting. With just the two cases, it is impossible to get a truly representative sample, but I am planning to do a much larger, systematic study in the future!

Section 3.3: I think it may be somewhat difficult/potentially problematic to combine the precip threshold analysis for the two cases given that there were microphysics changes between the two HRRR versions that likely affected these results.

This is a good point. I present the statistics separately in Figure 11, so the analysis itself is not together. I have added a caveat to the text about the model upgrade and I’ve made a note that broad conclusions cannot be drawn.

Line 619: clarify that this statement is valid for these two cases specifically

Good catch. I have updated the text to stress that there were important subtle differences in 2016 that likely contributed to the location errors.